# Application of a Decisional Capacity Assessment for Older Research Participants with Cognitive Impairment

**DOI:** 10.3390/bs13090767

**Published:** 2023-09-14

**Authors:** Ling Xu, Noelle L. Fields, Megan R. Westmore, Kathryn M. Daniel, Brooke A. Troutman

**Affiliations:** 1School of Social Work, The University of Texas at Arlington, Arlington, TX 76010, USA; noellefields@uta.edu (N.L.F.); mxw6594@mavs.uta.edu (M.R.W.); 2College of Nursing and Health Innovation, The University of Texas at Arlington, Arlington, TX 76010, USA; kdaniel@uta.edu; 3McDermott Library, United States Air Force Academy, Colorado Springs, CO 80840, USA; brooke.troutman@afacademy.af.edu

**Keywords:** assessment, decision-making capacity, older adults with cognitive impairment, UBACC

## Abstract

Decisional capacity assessment is important for older adult participants who have cognitive impairment. This paper reports the implementation of the University of California, San Diego Brief Assessment of Capacity to Consent (UBACC) and its potential for practice and research. Nine of the 10 items remained to use except for adapting the last item. Approximately 130 older adults with cognitive impairment completed the UBACC screening. Item-by-item descriptive statistics, exploratory factor analysis (EFA), group comparisons of each item, as well as total sum scores of the UBACC were conducted. Results showed that the items that were most often answered correctly included item #10 (participant will be paid), item #4 (study is voluntary), and item #5 (can withdraw at any time). Conversely, the items that were most often answered incorrectly included item #9 (not any benefit potentially), item #7 (potential risk or discomfort), and item #6 (tasks during participation). Respondents with mild cognitive impairment had higher correct answer rates than those with advanced cognitive impairment. The UBACC screening tool has relative utility for older participants with cognitive impairment.

## 1. Introduction

Persons with Alzheimer’s disease and related dementias (ADRD) experience not only cognitive and functional losses but also losses in the ability to make decisions, known as decision-making capacity [1]. Decisional capacity assessment is different from cognitive screening and is important for research participants with cognitive impairment. Researchers have made substantial progress in developing a model of capacity assessment. Vignette-based instruments, such as the Assessment of the Capacity to Consent to Treatment (ACCT) Vignette [2], the Competency Interview Schedule (CIS) [3], the Decision Assessment Measure (DAM) [4], the Hopemont Capacity Assessment Interview (HCAI) [5,6], Thinking Rationally about Treatment (TRAT) [7], etc., as well as the structured or semi-structured interviews, such as Aid to Capacity Evaluation (ACE) [8], the Capacity Assessment Tool (CAT) [9], and the MacArthur Competence Assessment Tool—Treatment (MacCAT-T) [10], etc., have been used for decisional capacity assessment; however, most of them are lengthy and hard to administer, especially for older adult participants with cognitive impairment [1]. These challenges often pose barriers to conducting studies with persons with ADRD, and additional strategies are needed to guide researchers involved in intervention research with this population.

The University of California, San Diego Brief Assessment of Capacity to Consent (UBACC) was developed and tested to help investigators identify research participants who warrant a more thorough decisional capacity assessment prior to enrollment [11]. The UBACC is a 10-item scale that includes questions focusing on understanding (4 items), appreciation (5 items), and reasoning (1 item) of the information concerning a research protocol. Every item receives a score ranging from 0 to 2 points, where ‘0’ represents a response that clearly lacks capability, and ‘2’ signifies a response that clearly demonstrates capability. If a response is only partially appropriate or if uncertainty remains even after re-explanation and additional probing, an intermediate score of ‘1’ can be assigned. The cut-off point (sum score ≥ 14.5) determines the participants’ capacity to make a decision. The UBACC is user-friendly, and it typically takes less than 5 min to administer by a bachelor’s degree-level researcher. This decision-making capacity scale has shown good internal consistency, interrater reliability, concurrent validity, high sensitivity, and acceptable specificity for healthy participants and those with mental illness [11], as well as for individuals contemplating participation in Alzheimer’s disease research [12]. We used this UBACC for a research project that has older adult research participants who have cognitive impairment. Participants’ engagement in completing the UBACC is a preliminary step to engaging in our research project. For further information regarding this research project, kindly consult our published work [13]. In this research report, we present how this assessment was used and what lessons we learned from this assessment.

## 2. Methods

### 2.1. Application of UBACC

The literature [11,12] suggests some items of the UBACC may not be relevant for certain types of studies, and it is recommended that researchers determine the essential items to be included in order to ensure valid consent for the specific protocol. For the present study, we adopted this scale to assess the decisional capacity of potential older participants who were interested in participating in a research project. Nine of the 10 items of the UBACC fit our current research protocol. However, the last one/item needed to be modified due to its irrelevance to the research project, which is a telephone-based social intervention (more information can be found in our published work [13]). We replaced the original last item, “Who will pay for your medical care if you are injured as a direct result of participating in this study?” with a question, “Will you be compensated/paid for your effort in participating in this study?”. This modification was approved by the institutional review board (IRB).

### 2.2. Procedure of UBACC Assessment

The present research study was a collaboration with Meals on Wheels (MOW) in Tarrant County, Texas. As per the approved IRB protocol, case managers from MOW first called the potential older adult participants who were eligible for the study (aged 65+, had cognitive impairment measured by the Ascertain Dementia-8 Item (AD8) questionnaire with AD8 ≥ 2) [14] and briefly discussed the research project. If a participant expressed interest in the study to the MOW case manager and granted permission for the research team to call them to explain the study in greater detail, two researchers called each of the potential older adult participants. On average, each researcher called approximately 65 older adults, asking the same 10 questions after having explained the study procedure to them. During the call, the researchers provided detailed information on the research protocol, mainly based on the consent form and a telephone script approved by the IRB. The researchers then asked the 10 questions of the UBACC to make sure that every participant understood the study and knew what they were volunteering to do. If an older adult had total scores of 14.5 or above, they were asked to verbally consent to participate after the researcher read the informed consent. If not, the researcher explained that they did not meet the inclusion criteria for the study and said thank you to the participant. Approximately 130 individuals completed the assessment using the UBACC. 

### 2.3. Ethical Considerations

The protocol and consent forms underwent approval by the University of Texas at Arlington (UTA) IRB prior to commencing the study procedures. Participants’ names were accessible solely to the research team. To ensure confidentiality, individual names were replaced with unique study identifiers or numbers. All data files were securely stored within encrypted, password-protected folders on laptops provided by the UTA. These laptops adhered to encryption standards outlined by the UTA Office of Information Technology.

Before taking part in the study, all participants provided informed consent. It was made clear to potential participants that their decision to decline participation or withdraw from the study would in no way affect their current or potential future services from MOW or the Area Agency on Aging. If requested, participants received a copy of the consent form for their records.

Regarding human subject risk, the study posed minimal harm. If any participant experienced emotional distress, Daniel, a Registered Nurse and Gerontological Nurse Practitioner, along with Fields, a licensed clinical social worker, were able to provide the necessary support and counseling to address any potential emotional distress. They were also able to facilitate access to resources through the local Area Agency on Aging for older adults, as needed.

### 2.4. Data Analyses

Univariate descriptive statistics (i.e., frequencies) were obtained for each of the UBACC scale’s ten items to assess which items were most or least likely to be answered correctly. Next, a reliability analysis was conducted on the scale as a whole, as well as an exploratory factor analysis (EFA) to compare the underlying factor structure of this scale to that of the original scale established by Jeste et al. [11]. Finally, bivariate analyses between cognitive impairment (measured by AD8) and capacity to consent (measured by the UBACC) were conducted. Group comparisons of each item, as well as total sum scores of the UBACC, were conducted between those who had minor (AD8 = 2) and more advanced (AD8 > 2) cognitive impairment using an independent samples *t*-test. All analyses were completed in SPSS for Windows (Version 28.0) [15].

## 3. Results

Among the 130 older adults who completed the UBACC screening, they ranged in age from 65 to 93, with an average age of 75 (*SD* = 7.5). Just over 80% of participants were female, with the remaining participants identifying as male. The items that were most often answered correctly included item #10 (98.5% correct), “Will you be compensated/paid for your effort in participating in this study?”; item #4 (96.2% correct) “Do you have to be in this study if you do not want to participate?”; and item #5 (94.6% correct), “If you withdraw from this study, will you still be able to receive regular service or other benefits you generally have?”. Conversely, items with the lowest percentage of correct responses included item #9 (12.3% correct), “Is it possible that being in this study will not have any benefit to you?”; item #7 (53.1% correct), “Please describe some of the risks or discomforts that people may experience if they participate in this study,”; and item #6 (72.3% correct), “If you participate in this study, what are some of the things that you will be asked to do (tasks)?”. See Table 1 for full descriptive results.

When looking at the UBACC as a scale, the Cronbach’s alpha was low (α = 0.30). The exploratory factor analysis (EFA) was then conducted. Before running the EFA, the Kaiser–Meyer–Olkin Measure of Sampling Adequacy was checked with 0.507, which was mediocre. The Bartlett’s measure tests (*χ*^2^ (45) = 70.83, *p* < 0.01) showed there were enough sizable correlations to justify conducting factor analysis. Originally, with Promax rotation, however, the correlation among factors was low (>0.2), so the analysis was rerun with Varimax rotation. The results of the EFA showed 4 common factors when retaining eigenvalues larger than 1, which explained 53.97% variances in total (see Table 2). Factor 1 includes items #1, #6, #7, and #10, which represent “project understanding”. Factor 2 has items #2 and #5, which is more of a “reason for participating.” Factor 3 (items #3 and #4) represents “voluntary participation,” and Factor 4 (items #8 and #9) indicates “benefits of participating”.

As shown in Table 3, the bivariate analyses showed significant group differences in the total scores of the UBACC between participants with minor or advanced cognitive impairment (*t* _(128)_ = 3.68, *p* < 0.001). Group comparison of the UBACC responses based on the level of cognitive impairment also showed that the percentage of respondents with mild cognitive impairment who answered an item correctly was higher than the percentage of respondents with more advanced cognitive impairment, except item #2 (“What makes you want to consider participating in this study?”) (90.2% vs. 92.1%). Results of the *t*-tests indicated a statistically significant group difference in the percentage of correct answers for item #1 (“What is the purpose of the study that was just described to you?”), item #3 (“Do you believe this is primarily research or primarily treatment?”), item #5 (“If you withdraw from this study, will you still be able to receive regular service or other benefits you generally have?”), and item #9 (“Is it possible that being in this study will not have any benefit to you?”). In addition, the average total UBACC score for participants with mild cognitive impairment was statistically significantly higher than for those with more advanced cognitive impairment. 

## 4. Discussion

It is critical for researchers to screen and document the decisional capacity of participants who have cognitive impairment before the start of a research study. For our study, we adapted (minor change) the UBACC and used this to assess the decisional capacity of older adult participants with cognitive impairment. This study added to the limited literature on decisional capacity assessment for older adults with cognitive impairments, especially for applying the UBACC to such a population. The research team felt it was easy to administer and use the UBACC. In general, the assessment process and the results showed that the UBACC is a useful assessment tool for decisional capacity. 

When looking at the UBACC item by item, it was found that three items had the most frequent wrong answers. The most frequently misunderstood item was “Is it possible that being in this study will not have any benefit to you?” (87.7%). Even after the researchers re-explained the potential risk of there being no benefit at all from participating in the study, some older adults insisted that they would benefit from the study in some way. The second most frequently misunderstood item was “Please describe some of the risks or discomforts that people may experience if they participate in this study” (46.9%). When asking this question, the most common wrong answer was “no risks for me.” Similar to the most frequently misunderstood item, many older adult participants in this study stated this study would benefit them and that no risks or discomforts would occur to them. Several reasons might help explain this phenomenon. First, older adults were recruited from MOW, and they initially heard about this study from their case managers whom they trusted. Second, older adults felt very positively about our social study as it posed minimal risks to participants. Third, there is always the potential for slight differences when two researchers are conducting assessments, which may give rise to difficulties in understanding the questions for older adult participants. This may also help explain the many wrong answers to the question, “If you participate in this study, what are some of the things that you will be asked to do?” (22.3%). Older adults might be confused about the words “things you will be asked to do.” Therefore, clear and concise explanations or instructions, as well as easy-to-understand wording, are important in conducting the assessment.

On the other hand, the older adult participants in this study were clear about the purpose of the study, why they wanted to join, as well as the benefits, the voluntary nature, and the compensation for participating. These are important components of decision-making in whether to participate in the study as well as how to participate (steps in the participating process). Based on our observations, those older adults who passed the decisional capacity screening could well understand and participate in the research project. In addition, as our study shows in Table 3, older adults with more advanced cognitive impairment, in general, were less likely to have high scores in answering the UBACC decisional capacity screening items. They generally had higher rates of wrong answers on each of the 10 items compared to those with minor cognitive impairment. Therefore, these older participants with more advanced cognitive impairment may have a poorer understanding of the screening items or the research project. This finding is in line with existing literature, which suggests that the UBACC has concurrent validity with related constructs [12] and possesses sufficient sensitivity to detect impaired decisional capacity to avoid false-negative errors and specificity to avoid false-positive errors [11].

When looking at the UBACC as an assessment scale, this study did not show an acceptable reliability score. This is inconsistent with a previous study that showed that the UBACC had a 0.77 Cronbach score of internal consistency for the 127 patients with schizophrenia and a 0.76 Cronbach score among the 30 healthy subjects [11] as well as a 0.747 score for people participating in ADRD research [12]. In addition, the previous study by Jeste and colleagues [11] also showed three factors were generated with eigenvalues greater than 1, and they explained 56% of the variance. Seaman et al. [12] showed a one-factor solution with 8 items [12]. However, the present study showed 4 factors instead. Several factors may contribute to this inconsistency. The difficulty level and clarity of expression of a test item are potential crucial factors that can impact the reliability of test scores. When test items are either too easy or too difficult for the participants, it tends to result in lower reliability scores. In other words, if the items are not appropriately challenging or comprehensible for the test-takers, the reliability of the scores may be compromised [16]. The study by Jeste and colleagues [11] was conducted among participants with mental illness and healthy subjects, which are different from the participants in the present study, who had cognitive impairment. The 10 items in the UBACC might be harder for older adults with cognitive impairment to understand compared to those with mental illness. Another possible reason is the sample size. Kaiser–Meyer–Olkin is a measure of sampling adequacy. Its mediocre size in the present study might raise a concern about the inadequate sample size. 

This study has some limitations. First, although the UBACC seems to have face and content validity in the present study, as observed by the researchers, we did not validate the UBACC scale in other dimensions. Though the UBACC has shown good validity in the literature for healthy participants and those with mental illness [11,12], we could not examine its validity in the present study among older adult participants with cognitive impairments. However, our group analyses between participants with minor and advanced cognitive impairments suggest that the UBACC may have sufficient sensitivity as a screening tool for this population. Further studies may conduct convergent and discriminant validity. Second, the older adult participants were MOW clients who were low-income, home-bound, and community-dwelling older adults. Consequently, this sample was not representative and is subject to potential selection bias. 

Despite these limitations, the present brief report has some implications for practice. This UBACC decisional capacity screening tool has relative utility for older participants with cognitive impairment. However, researchers need to be careful in the screening process. For example, researchers conducting the screening need to make sure they cover all the contents of the 10 items that the screening test asks when explaining the research protocol to the participants. Researchers also need to be careful when communicating with participants who have cognitive impairment before and during the screening. For example, researchers need to use concise and clear wording when describing the research protocol and need to break questions and tasks down into multiple parts if needed [17]. In addition, research suggests that memory-impaired individuals can participate in survey research if the instrument is simple, has dichotomous response categories, and contains unidirectional frequency/amount items when possible [18]. Therefore, researchers should follow these rules when asking the 10 items of the UBACC to screen participants with cognitive impairment. When utilizing the UBACC in a particular study, Jeste et al. [11] suggest that investigators should determine the essential items among the 10 included in the assessment. This is necessary to ensure valid consent for the specific protocol, as certain items may not be relevant to certain types of studies. Additionally, researchers should identify the specific responses that warrant full credit (2 points) on these essential items. By doing so, the UBACC can be seamlessly incorporated into the consent process of most clinical research projects involving participants with or without cognitive impairment. This is especially valuable in situations where a portion of potential participants may be expected to have insufficient understanding or appreciation of the provided study information.

## 5. Conclusions

Assessing decisional capacity holds significance for older adult participants with cognitive impairment. In our project, we employed the UBACC to evaluate the decisional capacity of older adults with cognitive impairment. This assessment served as one of the criteria for participation eligibility. The UBACC proved to be user-friendly for our research team. Descriptive statistical analysis of the results highlighted certain trends. Notably, respondents were more accurate in answering items related to the research’s voluntary nature (e.g., can withdraw at any time), while their accuracy was comparatively lower in relation to the project’s benefits or risks (e.g., will benefit no matter what, or no risks identified). It was also observed that individuals with mild cognitive impairment exhibited higher rates of correct responses compared to those with more advanced cognitive impairment. In summary, the findings from this brief report indicate that the UBACC screening tool holds relative utility for older participants with cognitive impairment.

## Figures and Tables

**Table 1 behavsci-13-00767-t001:** Descriptive statistics for adapted UBACC items (*N* = 130).

UBACC Items	% Correct	% Incorrect
1. What is the purpose of the study that was just described to you?	89.2	10.8
2. What makes you want to consider participating in this study?	90.8	9.2
3. Do you believe this is primarily research or primarily treatment?	77.7	22.3
4. Do you have to be in this study if you do not want to participate?	96.2	3.8
5. If you withdraw from this study, will you still be able to receive regular service or other benefits you generally have?	94.6	5.4
6. If you participate in this study, what are some of the things that you will be asked to do?	72.3	27.7
7. Please describe some of the risks or discomforts that people may experience if they participate in this study.	53.1	46.9
8. Please describe some of the possible benefits of this study.	90.8	9.2
9. Is it possible that being in this study will not have any benefit to you?	12.3	87.7
10. Will you be compensated/paid for your effort in participating in this study?	98.5	1.5

**Table 2 behavsci-13-00767-t002:** Exploratory factor analyses of UBACC items (*N* = 130).

UBACC Items	Factor 1	Factor 2	Factor 3	Factor 4
1. What is the purpose of the study that was just described to you?	0.625			
2. What makes you want to consider participating in this study?		0.657		
3. Do you believe this is primarily research or primarily treatment?			0.711	
4. Do you have to be in this study if you do not want to participate?			0.639	
5. If you withdraw from this study, will you still be able to receive regular service or other benefits you generally have?		0.805		
6. If you participate in this study, what are some of the things that you will be asked to do?	0.697			
7. Please describe some of the risks or discomforts that people may experience if they participate in this study.	0.552			
8. Please describe some of the possible benefits of this study.				0.590
9. Is it possible that being in this study will not have any benefit to you?				−0.660
10. Will you be compensated/paid for your effort in participating in this study?	−0.410			

**Table 3 behavsci-13-00767-t003:** Percentage of respondents correctly answering UBACC items by level of cognitive impairment (*N* = 130).

UBACC Items	Mild *n* (%)	More Advanced *n* (%)	Comparison *t*-Test
1. What is the purpose of the study that was just described to you?	84 (91.3)	32 (84.2)	−1.06 *
2. What makes you want to consider participating in this study?	83 (90.2)	35 (92.1)	0.35
3. Do you believe this is primarily research or primarily treatment?	77 (83.7)	24 (63.2)	−2.33 ***
4. Do you have to be in this study if you do not want to participate?	89 (96.7)	36 (94.7)	−0.49
5. If you withdraw from this study, will you still be able to receive regular service or other benefits you generally have?	90 (97.8)	33 (86.8)	−1.91 ***
6. If you participate in this study, what are some of the things that you will be asked to do?	69 (75.0)	25 (65.8)	−1.02
7. Please describe some of the risks or discomforts that people may experience if they participate in this study.	53 (57.6)	16 (42.1)	−1.61
8. Please describe some of the possible benefits of this study.	85 (92.4)	33 (86.8)	−0.89
9. Is it possible that being in this study will not have any benefit to you?	13 (14.1)	3 (7.9)	−1.09 *
10. Will you be compensated/paid for your effort in participating in this study?	91 (98.9)	37 (97.4)	−0.54
*Mean* total score (*SD*)	17.91 (1.15)	16.97 (1.66)	3.17 ***

* *p* < 0.05, *** *p* < 0.001. Two-tailed, equal variances are not assumed.

## Data Availability

The data presented in this study are available on request from the corresponding author.

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
