# Peer review of "Application of a Decisional Capacity Assessment for Older Research Participants with Cognitive Impairment"

_behavsci, 2023, doi:10.3390/bs13090767_

Round 1
Reviewer 1 Report
In fact, I think the paper addresses a really important topic, and the reviewers and I agree that there are merits. However, there are also some concerns.
The knowledge gap and purpose of the study are unclear.
The explanation of why the researchers only used 9 out of ten items of the tool and why they modified the last item needs to be mentioned.
How were the potential participants determined?
I think we need to know more about the sample. Is this sample representative? Is there a sample selection bias?
Were both researchers collected the same data from each participant or asked the same questions?
Did you assess the interrater reliability?
Why did you include the score on the UBACC as an inclusion criterion?
How was the sample size determined? Is it enough for significant statistical analysis?
Where is the full description of the tool in the measurement section?
the analysis of the tool’s validity is missing.
In general, the methods section lacks clarity. Please include ethical aspects considered.
The limitations and implications of the study are missing.
The conclusion must include the summary of the findings and inclusive statements but I see that you cited new references in the conclusion.
Minor editing of English language required.
Author Response
In fact, I think the paper addresses a really important topic, and the reviewers and I agree that there are merits. However, there are also some concerns.
- The knowledge gap and purpose of the study are unclear.
Response: Thank you for the opportunity to revise. Following the sentence toward the end of the initial paragraph (“however, most of them are lengthy and hard to administer, especially for older adult participants with cognitive impairment”), we added an additional sentence. “These challenges often pose barriers for conducting studies with persons with ADRD and additional strategies are needed to guide researchers involved in intervention research with this population.” We hope that this additional sentence clarifies the knowledge gap.
Additionally, the purpose of the study is articulated at the conclusion of the second paragraph, “In this research report, we presented how this assessment was used and what lessons we learned from this assessment”. Thus, our intent with our study is to illustrate the utilization of this assessment and the insights garnered from it.
- The explanation of why the researchers only used 9 out of ten items of the tool and why they modified the last item needs to be mentioned.
Response: The rational of minorly modifying the existing UBACC scale was stated on page 2, as: “Nine of the 10 items of UBACC fit our current research protocol, however, the last one/item needed to be modified due to its irrelevance to the research project, which is a telephone based social intervention (more information can be found in our published work). We replaced the original last item “who will pay for your medical care if you are injured as a direct result of participating in this study” with statement of “Will you be compensated/paid for your effort in participating this study?”. This modification was approved by the institutional review board (IRB).” Additionally, Jeste et al. suggest some items may not be relevant for certain types of studies and they recommend that researchers determine the essential items to be included in order to ensure valid consent for the specific protocol. We added this sentence to section 2.1 on page 2 in the revised manuscript.
- How were the potential participants determined?
Response: Once potential participants were read the research protocol, successfully completed the decisional capacity assessment, and maintained their interest in participation, they were then asked to voluntarily take part in the study and verbally consent to the study after reading the IRB approved form.
- I think we need to know more about the sample. Is this sample representative? Is there a sample selection bias?
Response: The potential participants were referred by Meals on Wheels (see sampling description on page 2). Therefore, they are not representative. There may have been some sample selection bias. We added this to the limitation section on page 6.
- Were both researchers collected the same data from each participant or asked the same questions?
Response: Both researchers asked the same questions for different participants. On average, each researcher called 65 older adults asking the same questions. To avoid confusion, we added a sentence on page 2.
- Did you assess the interrater reliability?
Response: This decision-making capacity scale has shown good internal consistency, interrater reliability, concurrent validity, high sensitivity, and acceptable specificity in literature (Jeste et al., 2007). In our study, we did not assess its interrater reliability.
- Jeste, D.V.; Palmer, B.W.; Appelbaum, P.S.; Golshan, S.; Glorioso, D.; Dunn, L.B.; Kim, K.; Meeks, T.; Kraemer, H.C. A new brief instrument for assessing decisional capacity for clinical research. Archives of General Psychiatry. 2007, 64(8), 966-974. https://doi.org/10.1001/archpsyc.64.8.966.
- Why did you include the score on the UBACC as an inclusion criterion?
Response: Incorporating decisional capacity as an inclusion criterion for participation serves to ensure that individuals involved in the study possess the cognitive ability to provide informed consent. This step helps to safeguard the ethical principles of autonomy and voluntariness, as participants with adequate decisional capacity are better equipped to comprehend the study's purpose, procedures, potential risks, and benefits. By setting decisional capacity as an inclusion criterion, the study aims to uphold the participants' ability to make informed choices and minimize the potential for coercion or exploitation.
- How was the sample size determined? Is it enough for significant statistical analysis?
Response: Power analyses performed with G*Power 3.1.9 indicated that a total of 92 dyads (or 184 participants) are required to address our research objectives using a linear mixed model. This sample size estimate was based on a small effect size (f =.22), alpha of .05, beta of .20, anticipated r over time of .30, 5% attrition rate based on the findings of Chung (2005), and two-tailed. As we integrated the UBACC tool to assess decisional capacity as a prerequisite for study participation, our sample size using this UBACC (n =130) surpasses the required number (n = 92). To access further information regarding this research project, kindly consult our published work (Xu et al., 2023). We have this information on page 2 in this brief report.
- Chung JC. An intergenerational reminiscence programme for older adults with early dementia and youth volunteers: values and challenges. Scandinavian journal of caring sciences. 2009;23(2):259-264.
- Xu, L.; Fields, N.L.; Cassidy, J.; Daniel, K.M.; Cipher, D.J.; Troutman, B.A. Attitudes toward aging among college students: Results from an intergenerational reminiscence project. Behavioral Sciences. 2023,13, 538. https://doi.org/10.3390/bs13070538.
- Xu, L.; Fields, N.L.; Daniel K.M.; Cipher, D.J.; Troutman, B.A. Reminiscence and digital storytelling to improve social and emotional well-being of older adults with ADRD: Protocol of mixed methods study design and a randomized control trial. JMIR Research Protocol. 2023. doi: 10.2196/preprints.49752.
- Where is the full description of the tool in the measurement section?
Response: The full description of the tool was presented under “Introduction” (lines 37-53). To avoid redundancy, we did not have the description of this tool in the method section. Instead, we provided description of “Procedure of UBACC Assessment”, where we described how we applied this tool assessment in our current study.
- The analysis of the tool’s validity is missing。
Response: The San Diego Brief Assessment of Capacity to Consent (UBACC) scale has been tested as reliable and valid in assessing participant’s capacity to make a decision in literature. For this brief report, we just described how we applied this scale to a unique population and lessons learnt when using it to older adults with cognitive impairment/s. We acknowledged this limitation in the discussion part on page 6.
- In general, the methods section lacks clarity. Please include ethical aspects considered.
Response: We added a section of 2.3 regarding ethical consideration on pages 2 and 3.
- The limitations and implications of the study are missing.
Response: We added limitations and implications on pages 6-7 as suggested.
- The conclusion must include the summary of the findings and inclusive statements but I see that you cited new references in the conclusion
Response: We removed the contents in conclusion part to study implications, and revised the conclusion part to include the summary of the findings and inclusive statements.
- Minor editing of English language required.
Response: Thank you for your comment. We have thoroughly reviewed the manuscript to ensure the absence of any grammar or spelling errors.
Reviewer 2 Report
1. Table 1 is unnecessarily convoluted when the authors switch between mentioning items with the “highest percentage of correct” and “highest percentage of incorrect” responses. The presentation in the table and corresponding text (lines 96 – 105) would be clearer if the authors consistently presented results on the percentage of correct responses since the percentage of incorrect responses is just the inverse (i.e., the data in the second column are redundant). It would help if the text describing the results in Table 1 mentioned items with the highest percentage of correct responses, followed by items with the lowest percentage of correct responses (not the highest percentage of incorrect responses).
2. The exploratory data analysis could use some more interpretation to help the reader understand what each factor might represent. Otherwise, delete the factor analysis results since they don’t seem to add much.
3. The authors need to double-check the significance level of the t-scores. There are well-defined cut-off points for a t-score that is significant at the .05, .01, and .001 levels. The significance levels indicated in this table are not correct.
4. Lines 175 – 178. The authors need to be careful using terms such as sensitivity and specificity, which are well-defined. The authors did not explicitly assess sensitivity or specificity.
Author Response
- Table 1 is unnecessarily convoluted when the authors switch between mentioning items with the “highest percentage of correct” and “highest percentage of incorrect” responses. The presentation in the table and corresponding text (lines 96 – 105) would be clearer if the authors consistently presented results on the percentage of correct responses since the percentage of incorrect responses is just the inverse (i.e., the data in the second column are redundant). It would help if the text describing the results in Table 1 mentioned items with the highest percentage of correct responses, followed by items with the lowest percentage of correct responses (not the highest percentage of incorrect responses).
Response: We appreciate this comment. We revised the description of Table 1 with the lowest percentages of correct responses.
- The exploratory data analysis could use some more interpretation to help the reader understand what each factor might represent. Otherwise, delete the factor analysis results since they don’t seem to add much.
Response: We added three sentences on page 4 to interpret the 4 factors from exploratory data analyses.
- The authors need to double-check the significance level of the t-scores. There are well-defined cut-off points for a t-score that is significant at the .05, .01, and .001 levels.
Response: Thanks for this comment. We re-ran the analyses and confirmed that the results in Table 3 are correct. Levene’s test for equality of variances were first conducted. The Levene’s Test based on the mean showed that the variances in UBACC items between people with mild and more advanced cognitive impairment were significantly different. Therefore, the t scores reported in Table 3 were reported based on equal variances not assumed. We added notes at the bottom of Table 3 to make it clearer.
- Lines 175 – 178. The authors need to be careful using terms such as sensitivity and specificity, which are well-defined. The authors did not explicitly assess sensitivity or specificity.
Response: Thank you for this comment. We deleted the sensitivity and specificity sentence.
Round 2
Reviewer 1 Report
Thank you for your effort. Approved.
Nothing.